# TCMAgent: A Multi-Agent Framework for General Traditional Chinese Medicine

## Abstract

A central challenge in artificial intelligence is designing systems that replicate expert cognition in domains where decisions require holistic data synthesis and deliberative reasoning. While large language models (LLMs) have achieved remarkable progress, their monolithic and sequential architectures impose cognitive bottlenecks that limit their ability to reason over multi-modal evidence or resolve competing hypotheses. Recent advances in multi-agent frameworks provide a new paradigm for overcoming these limitations by distributing reasoning across specialized agents and enabling structured deliberation. We present TCMAGENT, a novel multi-agent architecture that operationalizes a distributed and reflective reasoning workflow. Our framework introduces two key innovations: (*i*) *parallel evidence synthesis*, where agents process heterogeneous inputs concurrently to form a unified representation, and (*ii*) a *collaborative deliberation module*, inspired by clinical peer review, in which agents adversarially refine hypotheses to surface trade-offs and converge on robust decisions. This process is further enhanced by an experiential reflection mechanism that learns from historical reasoning traces, enabling continual self-improvement. We validate TCMAGENT on a multi-modal clinical benchmark in Traditional Chinese Medicine (TCM), a canonical domain where expert-level reasoning requires holistic integration of patient data and careful negotiation of conflicting principles. Experiments demonstrate that TCMAGENT significantly outperforms strong LLM baselines in safety, coherence, and interpretability of treatment recommendations. These results provide the first empirical evidence that distributed, deliberative agentic architectures can overcome the cognitive bottlenecks of monolithic models, marking a step toward safer and more reliable AI in knowledge-intensive domains. We release our code for reproduction in the anonymous repository here: https://anonymous.4open.science/r/TCM-Agent-B5EF.

## 1 Introduction

A key frontier in artificial intelligence (AI) is developing systems that can replicate expert cognition in domains requiring holistic data synthesis and deliberative reasoning. Such domains include clinical medicine, legal analysis, and strategic planning, where decisions depend on integrating diverse evidence and resolving competing principles. Current Large Language Models (LLMs), while powerful, are fundamentally limited by their sequential and monolithic architectures (Van et al., 2024; Zheng et al., 2024; Xiao et al., 2024; Luo et al., 2025; Cui et al., 2025; Ran et al., 2025). These limitations make it difficult for LLMs to emulate the distributed, reflective workflows that underpin real expert judgment. Traditional Chinese Medicine (TCM) presents a canonical example of this challenge. As a medical system practiced for millennia, TCM's efficacy hinges on holistic diagnosis derived from heterogeneous patient data (Yue et al., 2024b; Ma et al., 2021; Wang et al., 2023). Its global relevance, especially for chronic and complex conditions, underscores the need for computational frameworks capable of mastering this form of reasoning (Zhang et al., 2023; Zhuang et al., 2025). Yet the core cognitive tasks of TCM—synthesizing multi-modal evidence and deliberating over conflicting therapeutic principles—remain beyond the reach of conventional AI architectures (Zhang et al., 2025).

Existing LLM-based approaches for TCM reflect this mismatch (Zhang et al., 2025; Wei et al., 2024). Most adopt a **conversational, sequential-input paradigm**, where reasoning unfolds through

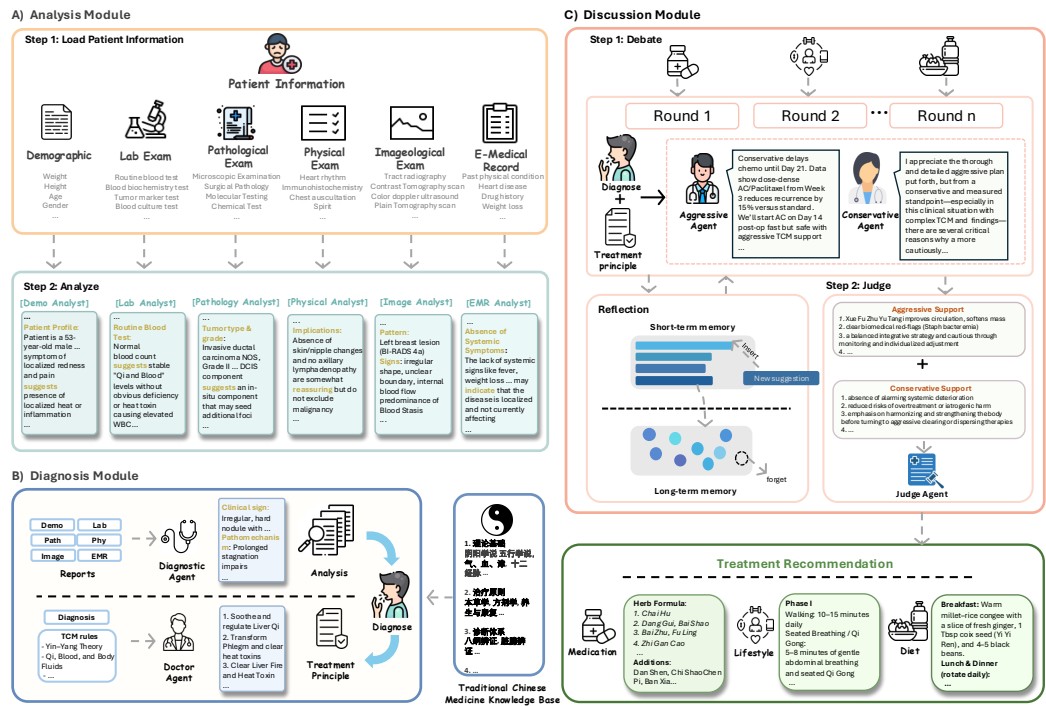

Figure 1: Overview of our proposed TCMAGENT framework. A) illustrates the analysis module, which take the input data and integrates the analysis. B) illustrates the diagnosis module. This module make the diagnose and design treatment principle with help of outer TCM knowledge base. C) illustrates the discussion module, where the debate happens. It incorporates a reflection module to help refine final treatment recommendation.

iterative prompt-response loops (Yang et al., 2025; Zhang et al., 2023; Zhuang et al., 2025; Chen et al., 2024). This fragments the clinical picture, preventing the holistic, single-pass assessments performed by human practitioners. As conversations scale, context is lost and models cannot autonomously synthesize a unified patient record, leading to inefficiency and diagnostic errors (Laban et al., 2025; Yang et al., 2025). Moreover, monolithic LLMs lack mechanisms for **structured, adversarial deliberation**, a cornerstone of sound judgment in high-stakes domains (Zhang et al., 2025; Liu et al., 2025; Wang et al., 2025a). Clinical decision-making is not a static classification task but a dynamic process of differential diagnosis, conflict resolution, and trade-off evaluation. Without a formal structure for proposing, critiquing, and refining hypotheses, LLMs are prone to plausible yet clinically flawed recommendations, a risk magnified by data scarcity in TCM (Yue et al., 2024b; Ren et al., 2022).

The recent maturation of multi-agent frameworks offers a new architectural paradigm to overcome these limitations. By distributing reasoning across specialized agents and enabling deliberative interaction, multi-agent systems make it possible to move beyond the sequential bottlenecks of monolithic models. To this end, we introduce TCMAGENT, a multi-agent framework that operationalizes the cognitive workflow of expert TCM practitioners. Unlike prior multi-agent systems that focus primarily on cooperative task decomposition or distributed planning, our framework introduces two previously unexplored innovations. First, we implement **parallel evidence synthesis**, where specialized agents concurrently process the entire multi-modal patient record to build a unified, holistic representation. Second, we design a **collaborative deliberation module** inspired by clinical peer review (Pedersen & Dyrkolbotn, 2014). Here, agents representing distinct clinical perspectives engage in a structured adversarial debate to challenge assumptions, surface trade-offs, and converge on robust conclusions (Arnesen et al., 2024; Nguyen et al., 2025). This process is further enhanced by an experiential reflection mechanism that allows the system to learn from its own reasoning traces, a form of meta-learning absent in conventional agent frameworks.

We empirically validate our framework through extensive experiments on a multi-modal TCM clinical dataset. Our results show that TCMAGENT consistently outperforms strong LLM baselines, sig-

nificantly improving the safety, correctness, and completeness of generated treatment plans across both proprietary and open-source models. Through systematic ablation studies, we isolate the impact of our core architectural innovations. We demonstrate that knowledge-grounded inference is critical for reducing factual errors and that the deliberative debate and reflection mechanisms are essential for surfacing clinical trade-offs and mitigating risks. These findings provide strong empirical evidence that by operationalizing a distributed and deliberative workflow, our framework successfully overcomes the cognitive bottlenecks of conventional, monolithic approaches.

**Our main contributions are as follows**: **(1)** We propose TCMAGENT, a novel multi-agent architecture that operationalizes a holistic and deliberative reasoning workflow, moving beyond the sequential processing limitations of current models. **(2)** We introduce a collaborative deliberation mechanism, where agents adversarially refine hypotheses to explicitly model clinical trade-offs, enhancing safety and interpretability. This is augmented by an experiential reflection module that enables continual self-improvement by learning from past reasoning traces and outcomes. **(3)** We provide the first empirical demonstration that a distributed and deliberative agentic architecture significantly outperforms monolithic LLM baselines in a complex, knowledge-intensive medical domain, validating its effectiveness in improving clinical safety, coherence, and interpretability.

## 2 RELATED WORK

**LLMs in TCM domain** LLMs have shown significant promise in field of TCM, addressing challenges such as the integration of holistic diagnostic approaches and the interpretation of complex medical texts (Chen et al., 2025; Kong et al., 2025). Recent studies demonstrate the potential of LLMs in various TCM tasks including clinical diagnosis and treatment recommendation (Haoyu et al., 2024; Yue et al., 2024b). Models such as `HuaTuoGPT` (Zhang et al., 2023; Chen et al., 2024), `JingFang` (Yang et al., 2025) and and `TCM-KLLaMA` (Zhuang et al., 2025) have been trained on huge volumes of TCM-specific corpora. These corpora are collected from real-world conversationss between doctor and patient and then turned into format of Question-Answering (QA) for purpose of instruction fine-tuning, achieving notable performance improvements (Zhang et al., 2023; Chen et al., 2024). Despite these advancements, challenges remain in adapting LLMs to TCM. Studies have pointed out issues such as data scarcity, model adaptability and the standardization of evaluation metrics (Zhang et al., 2025). Furthermore, limitation of conversation-based methods restricts model make a holistic decision given patient information (Laban et al., 2025; Liu et al., 2023). These limitations underscore the need for more sophisticated approaches to fully harness the potential of LLMs in the TCM domain.

**LLM Agent for medical decision-making** The integration of multi-agent systems with LLMs has been explored to address complex medical tasks that require specialized knowledge and collaborative reasoning (Wang et al., 2025a;b; Tang et al., 2023). One notable example is the `ClinicalAgent` system (Yue et al., 2024a), which utilizes a multi-agent architecture to perform clinical trial tasks. By combining `GPT-4` (Achiam et al., 2023) with multi-agent architectures and advanced reasoning technologies, `ClinicalAgent` demonstrates enhanced performance in clinical trial outcome prediction, achieving a noticeable improvement of predictive performance. Similiarly, Pandey et al. (2024) employs specialized LLM agents to automate prior authorization tasks in healthcare. Their system breaks down complex tasks into manageable sub-tasks, improving efficiency and accuracy in determining medical necessity. These studies highlight the effectiveness of multi-agent systems in handling intricate medical decision-making processes. However, the application of such systems to TCM remains limited, with few studies exploring the potential of multi-agent LLMs in this domain

## 3 THE TCMAGENT FRAMEWORK

### 3.1 PROBLEM FORMULATION

The clinical reasoning process in Traditional Chinese Medicine (TCM) is a complex, multi-step inference task over high-dimensional, heterogeneous data. We formalize this challenge as constructing a mapping function $F : \mathcal{P} \to \mathcal{T}$ from a multi-modal patient space $\mathcal{P}$ to a composite treatment space $\mathcal{T}$. For any given patient $i$, the input $p_i \in \mathcal{P}$ is a tuple of six modalities, $p_i = (p_i^{(1)}, \ldots, p_i^{(6)})$, where

each modality $p_i^{(k)}$ resides in a distinct data space $\mathcal{X}^{(k)}$. The input space is thus a Cartesian product, $\mathcal{P} = \prod_{k=1}^{6} \mathcal{X}^{(k)}$, encompassing demographics, laboratory results, pathology reports, physical examinations, imaging, and electronic medical records (EMRs).

The target output $\mathcal{T}_i \in \mathcal{T}$ is a structured tuple comprising three interdependent components: herbal medication ($\mathcal{T}_i^{\mathrm{med}}$), lifestyle interventions ($\mathcal{T}_i^{\mathrm{life}}$), and dietary guidelines ($\mathcal{T}_i^{\mathrm{diet}}$). The output space is defined as $\mathcal{T} = \mathcal{Y}^{\mathrm{med}} \times \mathcal{Y}^{\mathrm{life}} \times \mathcal{Y}^{\mathrm{diet}}$. Our objective is to design an autonomous system $F$ that can reliably generate a clinically valid and personalized recommendation $\mathcal{T}_i$ for any input $p_i$. The central architectural challenge is to structure $F$ such that it is not only context-aware for individual patients but also faithful to the deep, structured knowledge inherent in TCM.

## 3.2 Architectural Overview

We introduce TCMAGENT, a multi-agent framework that operationalizes the clinical workflow as a compositional sequence of three specialized modules: Distributed State Encoding ($F_{\mathrm{enc}}$), Knowledge-Grounded Inference ($F_{\mathrm{inf}}$), and Deliberative Recommendation Generation ($F_{\mathrm{rec}}$). The global mapping $F$ is thus a functional composition:

$$F = F_{\mathrm{rec}} \circ F_{\mathrm{inf}} \circ F_{\mathrm{enc}}.$$

First, the *Distributed State Encoding* module leverages parallel agents to transform the raw, multi-modal input $p_i$ into a unified patient representation $\mathcal{R}_i$. Next, the *Knowledge-Grounded Inference* module anchors this representation in a domain-specific knowledge base to produce a formal diagnosis $\mathcal{D}_i$ and a guiding treatment principle $\mathcal{P}_i$. Finally, the *Deliberative Recommendation Generation* module simulates a structured, adversarial debate to synthesize this information, weigh clinical trade-offs, and generate the final recommendation $\mathcal{T}_i$.

## 3.3 Distributed State Encoding Module

To address the challenge of heterogeneous and high-dimensional inputs, the encoding module ($F_{\mathrm{enc}}$) parallelizes information processing. It employs a set of modality-specific encoder agents, $\{\mathcal{A}_{\mathrm{enc}}^{(k)}\}_{k=1}^{6}$, each responsible for a single data stream. Each agent $\mathcal{A}_{\mathrm{enc}}^{(k)} : \mathcal{X}^{(k)} \to \mathcal{R}^{(k)}$ functions as a domain expert, mapping its assigned modality into a structured report space $\mathcal{R}^{(k)}$. This process occurs concurrently across all modalities:

$$\forall k \in \{1, \ldots, 6\}, \quad \mathcal{R}_i^{(k)} = \mathcal{A}_{\mathrm{enc}}^{(k)}(p_i^{(k)}).$$

Each report $\mathcal{R}_i^{(k)}$ contains distilled observations and preliminary interpretations. These distributed analyses are subsequently integrated by an aggregation function $g_{\mathrm{agg}}$ into a holistic patient state representation $\mathcal{R}_i \in \mathcal{R}$:

$$\mathcal{R}_i = g_{\mathrm{agg}}(\mathcal{R}_i^{(1)}, \ldots, \mathcal{R}_i^{(6)}).$$

This architecture mitigates information bottlenecks and attentional bias associated with sequential processing, ensuring a robust foundation for downstream reasoning.

## 3.4 Knowledge-Grounded Inference Module

The inference module ($F_{\mathrm{inf}}$) translates the synthesized patient state $\mathcal{R}_i$ into an interpretable, clinically actionable context. It begins with a diagnostic agent, $\mathcal{A}_{\mathrm{diag}} : \mathcal{R} \to \mathcal{Y}^{\mathrm{diag}}$, which identifies the underlying TCM syndrome pattern, yielding a formal diagnosis $\mathcal{D}_i = \mathcal{A}_{\mathrm{diag}}(\mathcal{R}_i)$ that is explicitly traceable to evidence in $\mathcal{R}_i$.

To ensure reasoning is grounded in established medical knowledge, we employ a retrieval-augmented mechanism. The diagnosis $\mathcal{D}_i$ serves as a query to a vectorized TCM knowledge base $\mathcal{K}$ to retrieve the top-$k$ most relevant clinical precedents or guidelines, denoted $D_i^{\mathrm{ret}}$:

$$D_i^{\mathrm{ret}} = \operatorname*{topk}_{d_j \in \mathcal{K}} \left(\mathrm{sim}(\mathrm{embed}(\mathcal{D}_i), \mathrm{embed}(d_j))\right).$$

A principle agent, $\mathcal{A}_{\mathrm{princ}} : \mathcal{Y}^{\mathrm{diag}} \times \mathcal{K}^k \to \mathcal{Y}^{\mathrm{princ}}$, then synthesizes the diagnosis $\mathcal{D}_i$ and the retrieved documents $D_i^{\mathrm{ret}}$ to formulate a high-level therapeutic principle $\mathcal{P}_i$. The module's output is a unified clinical context tuple, $C_i = (\mathcal{D}_i, \mathcal{P}_i)$, which serves as the scaffold for the final recommendation.

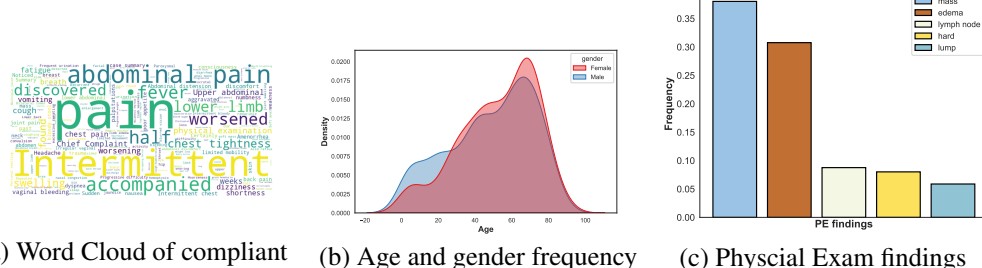

(a) Word Cloud of compliant    (b) Age and gender frequency    (c) Physcial Exam findings

Figure 2: **Data Descriptions**. (a) summarize the compliant of all patients. (b) shows the frequency of patients across different gender and age. (c) demostrates the Top-5 frequent Physical Exam findings.

### 3.5 DELIBERATIVE RECOMMENDATION GENERATION MODULE

The final recommendation is produced by the recommendation module ($F_{rec}$) through a policy of structured adversarial debate and case-based reasoning. We instantiate a debate between two agents with opposing clinical dispositions: an "aggressive" agent ($\mathcal{A}_{agg}$) that proposes assertive interventions and a "conservative" agent ($\mathcal{A}_{con}$) that raises cautionary counterpoints and prioritizes safety. Over $n$ rounds, these agents generate a deliberation trace $H^{(n)} = \{(u^{(j)}_{agg}, u^{(j)}_{con})\}^n_{j=1}$, conditioned on the clinical context $C_i$. A neutral judge agent, $\mathcal{A}_{judge}$, resolves the debate by synthesizing the competing arguments into a final, balanced recommendation $\mathcal{T}_i$.

This adversarial process makes clinical trade-offs explicit, enhancing safety and interpretability. To further ground this reasoning, we incorporate a reflection mechanism that retrieves deliberation traces $\mathcal{H}^{hist}$ from historical cases with similar clinical contexts. This retrieved experience provides a contextual prior, enabling the judge to ensure consistency with established best practices:

$$\mathcal{T}_i = \mathcal{A}_{judge}(C_i, H^{(n)}, \mathcal{H}^{hist}).$$

By learning from a memory of past reasoning processes, the framework continually refines its decision-making policy, improving the reliability and robustness of its outputs.

## 4 EXPERIMENTS

### 4.1 SETUP.

**Dataset** In this study, we leverage data from **ClinicalLab** (Yan et al., 2024) with their permissions, which contains 1,500 examples with features including patients' medical histories, laboratory examinations, physical examinations, imaging studies, demographic information, and pathological assessments. Some statistical information of dataset is shown in Figure 2. Detail of dataset is shown in Appendix C.1.

**Baseline.** To assess the generality of our approach, we evaluate TCMAGENT across a diverse set of large language models (LLMs). The evaluation covers both proprietary models including `GPT-4.1-mini` (OpenAI, 2025), `Gemini-2.0-flash` (Comanici et al., 2025), and `Claude-3-haiku` (Anthropic, 2024)—and open-source models—such as `GPT-OSS-20b` (Agarwal et al., 2025), `LLaMA-3.3-70b-instruct` (Grattafiori et al., 2024), and `DeepSeek-v3` (Liu et al., 2024). In all cases, these LLMs also serve as the backbone for TCMAGENT, enabling a fair comparison across different model families. In terms of input, since our data consists of multi-dimension, we feed data to LLMs step-by-step in order to mimic clinical scenario, but for TCMAGENT, we can directly feed data of all dimensions.

**Evaluation.** We leverage LLM-AS-JUDGE (Gu et al., 2024) to evaluate outcomes due to data scarcity of evaluating agent framework in TCM domain. We adopt `GPT-4.1-mini` (OpenAI, 2025) with a sampling temperature set to 0, as the judge of evaluation. We use our designed metric to evaluate the treatment recommendations from multiple dimensions. The overview of designed metrics is shown in Appendix A.

Table 1: **Performance of TCMAGENT vs. Standalone LLMs.** Results are averaged across three recommendation types (Diet, Lifestyle, Medication). The highest score for each metric within a model pair is in **bold**. Metrics shown are Relevance (Rel.), Coherence (Coh.), Completeness (Complete.), Correctness (Cor.), Safety (Sa.), Complexity (Complex.), Actionability (Act.), and Terminology (Ter.).

| Model | Rel. ↑ | Coh. ↑ | Complete. ↑ | Cor. ↑ | Sa. ↑ | Complex. ↑ | Act. ↑ | Ter. ↑ |
|---|---|---|---|---|---|---|---|---|
| *Proprietary Models* | | | | | | | | |
| **GPT-4.1-mini** | | | | | | | | |
| w/o TCMAgent | 85.82 | 85.14 | 83.72 | 86.72 | 82.22 | 75.42 | 83.29 | 85.21 |
| w/ TCMAgent | **92.73** | **91.34** | **94.26** | **92.64** | **89.82** | **76.19** | **89.56** | **90.54** |
| **Claude-3-haiku** | | | | | | | | |
| w/o TCMAgent | 82.08 | 82.91 | 77.17 | 82.39 | 80.44 | 71.00 | 81.39 | 82.69 |
| w/ TCMAgent | **89.67** | **88.61** | **89.53** | **90.12** | **86.75** | **71.59** | **87.11** | **87.53** |
| **Gemini-2.0-flash** | | | | | | | | |
| w/o TCMAgent | 86.69 | **87.52** | 87.04 | **87.49** | 84.87 | **78.21** | 86.40 | **88.54** |
| w/ TCMAgent | **87.67** | 86.32 | **88.98** | 87.10 | **87.11** | 75.92 | **86.55** | 84.98 |
| *Open-Source Models* | | | | | | | | |
| **GPT-OSS-20b** | | | | | | | | |
| w/o TCMAgent | 80.01 | 79.31 | 79.32 | 80.97 | 83.11 | 50.81 | 74.58 | 80.17 |
| w/ TCMAgent | **91.80** | **89.68** | **92.28** | **91.73** | **92.53** | **50.93** | **88.45** | **90.80** |
| **LLaMA-3.3-70b** | | | | | | | | |
| w/o TCMAgent | **85.23** | **86.36** | 82.22 | 78.57 | **84.00** | **61.49** | **83.35** | 82.94 |
| w/ TCMAgent | 81.63 | 82.88 | **86.60** | **83.28** | 82.92 | 54.54 | 76.09 | **83.24** |
| **Deepseek-v3** | | | | | | | | |
| w/o TCMAgent | 80.84 | 80.12 | 78.85 | 82.29 | 82.44 | 58.37 | 74.95 | 82.93 |
| w/ TCMAgent | **93.38** | **91.47** | **93.42** | **93.91** | **94.50** | **61.81** | **89.28** | **94.13** |

## 4.2 MAIN RESULT

**TCMAgent against LLMs.** Table 1 provides a systematic comparison between closed-source and open-source LLMs with and without the integration of TCMAGENT. **First**, we find that equipping diverse backbone models with TCMAGENT generally leads to measurable performance gains across multiple evaluation dimensions, including relevance, coherence, completeness, and safety, though the scale of improvement varies depending on the model family. For example, in the case of strong proprietary backbones such as GPT-4.1-mini and Claude-3-haiku, incorporating TCMAGENT consistently elevates performance across nearly all criteria. This result is particularly noteworthy because it indicates that even highly capable commercial models can still benefit from the structured, multi-agent reasoning pipeline of TCMAGENT. In these settings, the framework contributes additional layers of interpretability and safety, thereby complementing the raw linguistic and generative capabilities of the backbone models.

**Second**, a different trend emerges in models like Gemini-2.0-flash, where the observed effect of integration is more nuanced. Specifically, while the framework significantly improves completeness and safety by enforcing knowledge-grounded reasoning and cross-agent validation, it introduces a slight decline in coherence and termination quality. This trade-off suggests that, for some already fluent and well-regularized models, the overhead of multi-agent debate and reflection may affect the naturalness or conciseness of generated outputs, even as factual reliability improves. Such a pattern points to the need for careful calibration between fluency-oriented and safety-oriented objectives when extending advanced commercial systems with multi-agent structures. **Moreover**, the case of LLaMA-3.3-70b highlights a unique and somewhat unexpected outcome. Here, while

completeness benefits from the structured workflow of TCMAGENT, other important metrics show noticeable declines. This discrepancy suggests that certain open-sourced backbones may not align well with the collaborative reasoning paradigm that TCMAGENT enforces, potentially due to differences in pretraining corpora, alignment strategies, or decoding dynamics. Such findings underscore the possibility that not all models derive uniform benefits from the same framework, and that mismatches between agent design and model architecture may lead to performance degradation in specific dimensions. The detailed result is shown in Section C.3

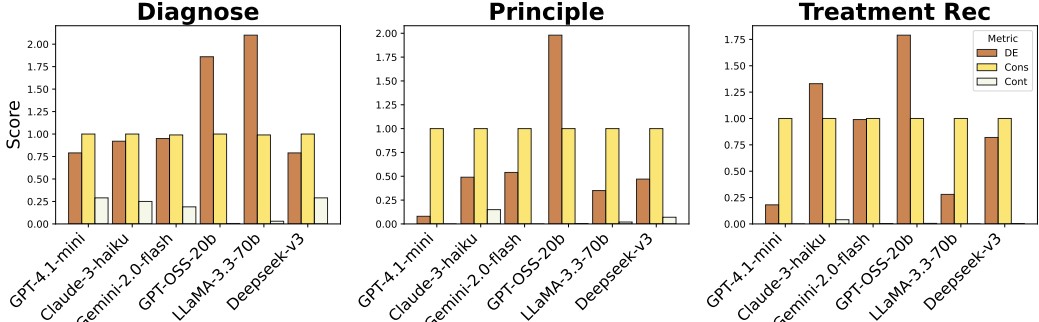

Figure 3: **Analysis result of TCMAGENT**. All models shown in x-axis are incorporated with TCMAGENT. Other than consistency, detected error and contraindication are the lower the better. The result of treatment recommendation is averaged from result of Diet, Lifestyle and Medication.

**TCMAgent analysis.** Figure 3 presents the performance of TCMAGENT under different LLM backbones. We have several observations from this figure: **(1)** In terms of consistency, all models remain at a stable level across diagnose, treatment principle and treatment recommendations. For example, consistency score of `GPT-4.1-mini` are close to one throughout three stages. It indicates the outputs of each module of TCMAGENT are logically connected with each other. **(2)** Contraindication tends to decrease from Diagnose stage to Treatment Recommendation stage, with the largest gains often appearing after the Principle stage

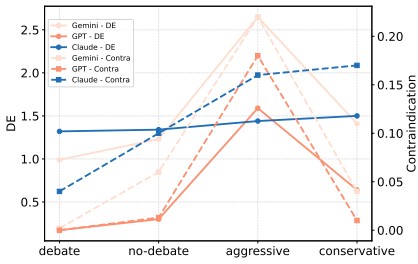

Figure 4: **Debate Analysis.** We evaluate the treatment recommendations and the result on the figure is averaged.

when safety checks are most explicit. It implies TCMAGENT increasingly enforces patient-specific constraints as the plan crystallizes, yielding safer recommendations. **(3)** The trend of detected error is fluctuating. For most models, the detected error would decrease in Principle stages compared to Diagnose stage, but the error would increase in Treatment recommendation stage. It reveals that within output from each module, there are some factual or logical errors. Such errors are hard to solve during the process of workflow moving forward. Taken together, these observations imply that TCMAGENT is capable of making patient safe. The logical connections between each module are strong while there are some errors within output of each module. The completed result is presented in Section C.3.

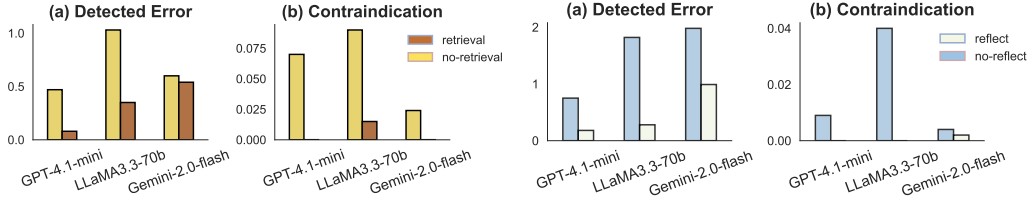

Figure 5: **Retrieval Analysis.** We evaluate the output of Treatment Principle.

Figure 6: **Reflection Analysis.** We evaluate the output of Treatment Principle.

**Impact of Retrieval** The impact of incorporating retrieval is presented in the left panel of Figure 5. We test the performance of TCMAGENT under backbone of `GPT-4.1-mini`, `LLaMA3.3-70b` and `Gemini-2.0-flash`. We observe that enabling retrieval leads to a substantial reduction in both Detected Error rate and Contraindication score. For example, the detected error and contraindication of `GPT-4.1-mini` drop from around 0.4 and 0.07 to around 0. It demonstrates that access to a structured Knowledge Base enhances the system's capacity to ground its reasoning in authoritative TCM knowledge. This grounding reduces the likelihood of generating factually incorrect or clinically unsafe treatment plans. Moreover, the observed improvement holds consistently across different backbone models of TCMAGENT, suggesting that the benefit stems from the retrieval mechanism itself rather than from the characteristics of a specific LLM. Detailed result is shown in Appendix C.3.

**Impact of Reflection** The effect of integrating the Reflection module is illustrated in the right panel of Figure 6. We test the performance of TCMAGENT under backbone of `GPT-4.1-mini`, `LLaMA3.3-70b` and `Gemini-2.0-flash`. We observe that incorporating reflection results in a clear reduction of both Detected Error and Contraindication scores, indicating that re-examining the debate process enables the system to identify and correct potential risks or inconsistencies that may otherwise remain unresolved. This reduction highlights that reflection contributes not only to mitigating local mistakes but also to strengthening the overall robustness of the reasoning trajectory. Furthermore, the consistent improvement observed across different backbone models demonstrates that the advantage of reflection is independent of any particular LLM architecture and instead derives from the structured self-assessment mechanism itself. Detailed result is shown in Appendix C.3.

**Impact of Debate** We investigate the effect of debate in Figure 4. We test the performance of TCMAGENT under backbone of `GPT-4.1-mini`, `Claude-3-haiku-20240307` and `Gemini-2.0-flash`. We study performance of TCMAGENT under setting of: (1) **With Debate**, (2) **No Debate**, (3) **Use opinion of Aggressive debator**, (4) **Use opinion of Conservative debator**. The overall trend is that debate has the lowest detected error. For example, `GPT-4.1-mini` achieve lowest detected error at debate. It suggests debate can help system avoid too risky treatment plan for patients. Also, we observe that no-debate achieves the second best performance in general. It indicates that the inherently the judge who is responsible for generating final treatment recommendation, is conservative about the treatment and thus less likely to voilate conservative principle to produce aggressive treatments, which would bring more

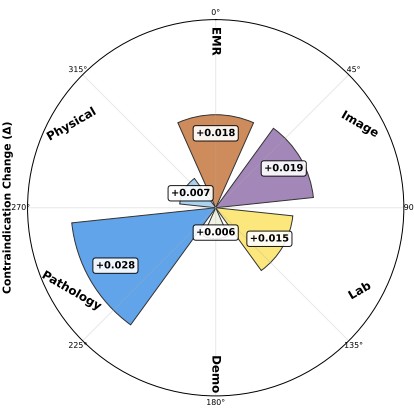

Figure 7: **Robust Analysis.** We evaluate TCMA-GENT with backbone of `Gemini-2.0-flash` under different missing input modality. The result is averaged. Lower the increase rate, lower sensitivity.

errors. Also we observe that unlike other two models, conservative debator performs worse than its aggressive debator. It suggests that the conservative debator in `Claude-3-haiku-20240307` may exhibit overly cautious tendencies, which paradoxically introduce errors by neglecting effective therapeutic options or by generating vague, underspecified recommendations. Such an outcome highlights that the benefit of debate does not stem solely from the presence of "conservative" voices, but from the dynamic interaction between divergent perspectives and the subsequent adjudication by the judge.

**Robustness study** We evaluate robustness of TCMAGENT under different missing input conditions and the result is shown in Figure 7. We investigate the contraindication change, where contraindication measures if treatment is risky to patient or not, of TCMAGENT with backbone of `Gemini-2.0-flash` under absence of each input feature (demographics, physical exam, lab exam, pathology, EMR, and imageological exam). In the figure, Lower the contraindication change,

less sensitivity for our framework toward the input feature. Results demonstrate that TCMAGENT exhibits differential sensitivity to various missing modalities. We observe that pathology and imageological examinations prove most critical for patient safety. Then, Laboratory examination and EMR data occupy intermediate positions. Demographic information shows the least impact. These findings imply availability of pathology and imageological exam are most insightful for diagnosis and treatment principle and thus can significantly improve reliability of treatment recommendations. Higher robustness to missing demographics suggests this information contributes less critically to safety-related decision making.

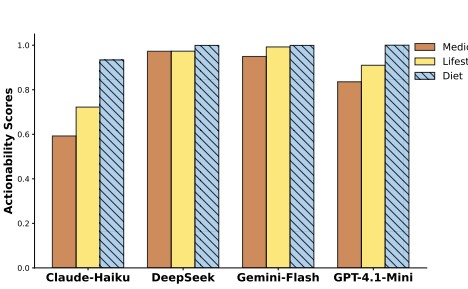

Figure 8: **Actionability Analysis.** We evaluate of actionability of TCMAGENT's treatment across different LLM backbones.

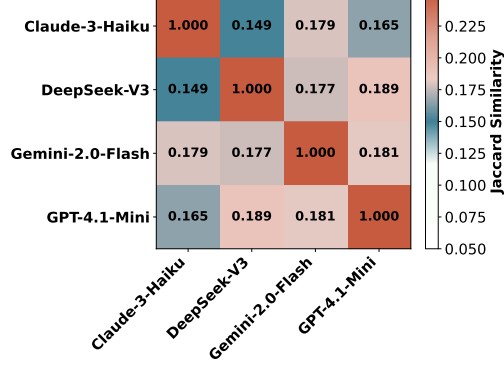

Figure 9: **Consistency Analysis.** We evaluate consistency of TCMAGENT's treatment recommendation with different LLM backbones.

**Cross-backbone Analysis** To evaluate cross-backbone consistency, which refers to the degree to which different backbones produce similar outputs when given the same patient case, we analyze semantic similarity using Jaccard similarity coefficients with TGMAGENT with backbones of four leading language models: `Claude-3-Haiku`, `DeepSeek-V3`, `Gemini-2.0-Flash`, and `GPT-4.1-Mini`. The result is presented in Figure 9. The range of similarity score is from 0.14 to 0.19, which exhibits a weak alignment. This pattern indicates that TCMAgent with different backbones display diversity when it comes to treatment recommendations, which is similar to real-world clinical scenario. Steps to calculate similarity score are presented in Section C.2

We further study the actionability of treatment across different backbones in Figure 8. Actionability refers to the degree to which treatment recommendations are expressed with concrete, executable. A consistent pattern appears across models: Diet recommendations achieve the highest actionability, reflecting that backbones often include concrete food suggestions and prohibitions. Lifestyle occupies the middle ground, where frequency and duration are sometimes specified but not as consistently detailed. Medication shows the lowest scores, indicating that although herbal prescriptions are proposed, they often lack precise dosage or administration instructions. This pattern suggests that the relative executability of recommendations is a systemic trend rather than model-specific noise. This implies that while TCMAGENT reliably generates actionable dietary and lifestyle guidance across backbones, further refinement is needed to enhance the precision of medication-level outputs, which remain weak in terms of clinical executability. Additional settings and analyses are provided in Section C.2.

## 5 CONCLUSION

In this work, we introduce TCMAGENT a novel multi-agent system designed to advance clinical decision-making in TCM. We highlight the challenges and limitations of current methods for TCM clinical decision-making task. Our proposed framework leverages multi-agent system, featured with debate mechanism, TCM knowledge retrieval and refection, to enhance TCM decision-making. We believe TCMAGENT paves the way for research of more intelligent and efficient system in TCM domain in the future.

## 6 ETHICS STATEMENT

This work adheres to the ICLR Code of Ethics, ensuring ethical compliance throughout all stages of the research. We contact authors of **ClinicalLab** (Yan et al., 2024) for obtaining permission and access to their dataset. We agree to their obtain the license right by agreeing restrictions that protecting patient pravicy, prohibition of tracking patient's individual information, do not distribute the data, and only use this dataset as research purpose.

## 7 REPRODUCIBILITY STATEMENT

To ensure research reproducibility, both the code and models are made available through our GitHub repository. The code used in this study will also be included as supplementary material. Detailed descriptions of the experimental settings are provided in Section 4 and Appendix C.3. A case study example and the prompts employed in this framework are presented in Appendix D and Appendix E, respectively.

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

## A  METRICS OVERVIEW

Here is a full list of designed metrics we use in this work. In total, we design 11 metrics, as shown in table 2. From top to bottom of the table, the first 8 metrics are used to compare treatment recommendations of TCMAGENT with those of baseline LLMs. The remaining 3 metrics are used to evaluate diagnosis, treatment principle and treatment recommendation generated by TCMAGENT.

| Metrics | Description |
| --- | --- |
| Coherence | Clarity, logical flow and ease of understanding |
| Relevance | How well suggestions address patient's condition and needs |
| Completeness | How comprehensive and detailed the recommendations are |
| Correctness | Alignment with Traditional Chinese Medicine principles and knowledge |
| Safety | Dosage safety of recommendation |
| Complexity | Challenges arising from a patient taking multiple suggestions |
| Actionability | How well the recommendation is executable |
| Terminology | Precision of terminology in recommendation |
| Detected Error | Factual or logical errors |
| Consistency | Logical connection with the previous module's response |
| Contraindication | Avoidance of patient-specific contraindications |

Table 2: Overview of evaluation metrics.

Followings are more detailed descriptions:

**Coherence** (Coh): ranging from 0 to 100 and higher the better, evaluates the clarity, logical flow, and readability of treatment recommendations, ensuring they are well-structured and easy to understand.

**Relevance** (Rel): ranging from 0 to 100 and higher the better, evaluates how well treatment recommendations are tailored to the patient's condition and individual needs, ensuring they are clinically appropriate.

**Completeness** (Complete): ranging from 0 to 100 and higher the better, evaluates how comprehensive and detailed the treatment recommendations are.

**Correctness** (Cor): ranging from 0 to 100 and higher the better, evaluates how well treatment recommendations adhere to TCM principles and knowledge, ensuring that prescriptions are appropriate, safe, and consistent with classical theory.

**Safety** (Sa): ranging from 0 to 100 and higher the better, evaluates whether treatment recommendations are dosage-safe and minimize risk to the patient.

**Complexity** (Complex): ranging from 0 to 100 and higher the better, evaluates the challenges and burdens a patient faces when following multiple treatment suggestions simultaneously.

**Actionability** (Act): ranging from 0 to 100 and higher the better, evaluates how practical and feasible the treatment recommendations are for patients, considering regimen complexity, clarity, and ease of adherence.

**Terminology** (Ter): ranging from 0 to 100 and higher the better, evaluates the accuracy, precision, and appropriateness of the language and terms used in treatment recommendations.

**Detected Error** (DE): ranging from 0 to $\infty$ and lower the better, evaluate the number of factual or logical errors.

**Consistency** (Cons): ranging from 0 to 1 and higher the better, evaluate how well the current module's response logically connects with and is consistent with the response from the previous module.

**Contraindication** (Cont): ranging from 0 to 1 and lower the better, evaluate how well the agent avoids violating patient-specific contraindications.

## B  IMPLEMENTATION DETAILS

We mainly use LLM API (OpenAI, Anthropic and etc.) to develop TCMAGENT. The TCMA-GENT system implements a multi-agent LLM framework using *LangGraph* orchestration, where the core `TCMAgentsGraph` class coordinates specialized agents through a state-driven workflow for Traditional Chinese Medicine diagnosis and treatment planning. The architecture features modular analyst agents for different medical data types a central diagnostic synthesis agent, and adversarial debate agents for balanced treatment recommendations across medication, diet, and lifestyle plans. Data flows through structured interfaces that process multi-modal patient information from JSON datasets, while maintaining context through AgentState and TreatmentDebateState objects with comprehensive logging for reproducibility.

## C  EXPERIMENT DETAILS

### C.1  DATASET DETAILS

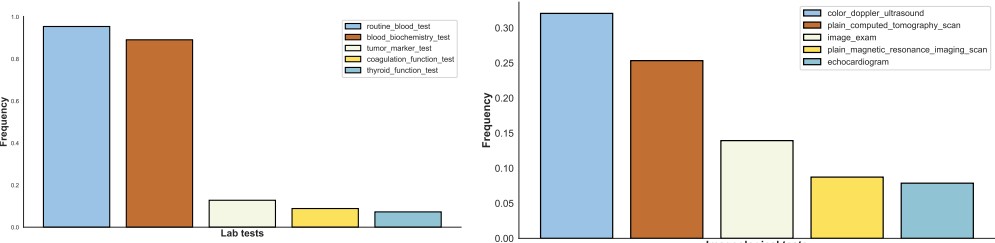

Figure 10: Top-5 frequent Lab test item

Figure 11: Top-5 frequent Imageological test item

In TCM clinical practice, practitioners develop standardized diagnostic workflows based on their expertise and accumulated experience (SUI et al., 2022; Ren et al., 2022; Matos et al., 2021). These workflows typically comprise four components: `Inspection`, `Auscultation and Olfaction`, `Inquiry`, and `Palpation`. Inspection involves evaluating the patient through visual observation of specific body parts to detect pathological changes. Auscultation and Olfaction encompass the assessment of body sounds and odors for diagnostic purposes. Inquiry refers to the process of eliciting the patient's subjective physiological and symptomatic experiences. Palpation is performed using a belt-like pressure sensor and amplifier attached to the wrist to dynamically measure the pulse waveform. Collectively, these procedures aim to provide practitioners with a comprehensive understanding of the patient's physiological condition. However, due to data format and scarcity issue, we use a different diagnostic structure.

We use dataset from **ClinicalLab** (Yan et al., 2024) with permission to evaluate our proposed framework and baselines. It contains 1,500 patient records. Following figures visualize some information in the dataset. Figure 10 and Figure 11 showcase the Top-5 frequent tested item by lab exam and imageological exam separately.

### C.2  DETAILED EXPERIMENT SETTING

**Cross-backbone Consistency.**  For our Traditional Chinese Medicine agent evaluation, we apply Jaccard similarity to measure agreement between different backbone's treatment recommendations by extracting treatment elements from each language model's output (medication plans, dietary recommendations, and lifestyle modifications), tokenizing these recommendations into sets of keywords, then calculating pairwise consistency with Jaccard similarity. For models $\mathcal{M} = \{Claude, DeepSeek, Gemini, GPT\}$ and case $i$, we calculate consistency as $\text{Consistency}_i =$

Table 3: Actionability assessment results for TCM treatment recommendations. Higher scores indicate greater actionability.

| Model | Medication | Diet | Lifestyle | Overall |
|-------|-----------|------|-----------|---------|
| Claude-3-Haiku | 0.592 | 0.934 | 0.722 | 0.749 |
| DeepSeek-v3 | 0.973 | 0.999 | 0.973 | 0.982 |
| Gemini-2.0-Flash | 0.949 | 0.999 | 0.992 | 0.980 |
| GPT-4.1-Mini | 0.836 | 1.000 | 0.910 | 0.915 |

$\frac{1}{\binom{|\mathcal{M}|}{2}} \sum_{m_1, m_2 \in \mathcal{M}, m_1 \neq m_2} J(R_{m_1}^{(i)}, R_{m_2}^{(i)})$, where $R_m^{(i)}$ represents the set of recommendations from model $m$ for case $i$.

**Cross-backbone Actionability.** We quantify the actionability of AI-generated TCM recommendations using a pattern-based slot-filling approach across three treatment domains: medication plans, dietary recommendations, and lifestyle modifications. For medication actionability ($A_M$), we detect dosage specifications and usage instructions, scoring as $A_M = (S_{\text{dose}} + S_{\text{usage}})/2$. Dietary actionability ($A_D$) identifies specific food items and restrictions, computed as $A_D = (S_{\text{foods}} + S_{\text{restrict}})/2$. Lifestyle actionability ($A_L$) extracts frequency ,duration , and timing information , calculated as $A_L = (S_{\text{freq}} + S_{\text{duration}} + S_{\text{timing}})/3$. The overall actionability score is Actionability $= (A_M + A_D + A_L)/3$, where each slot $S_i \in \{0, 1\}$ indicates binary presence of actionable information, yielding scores in $[0, 1]$ with higher values indicating more implementable recommendations.

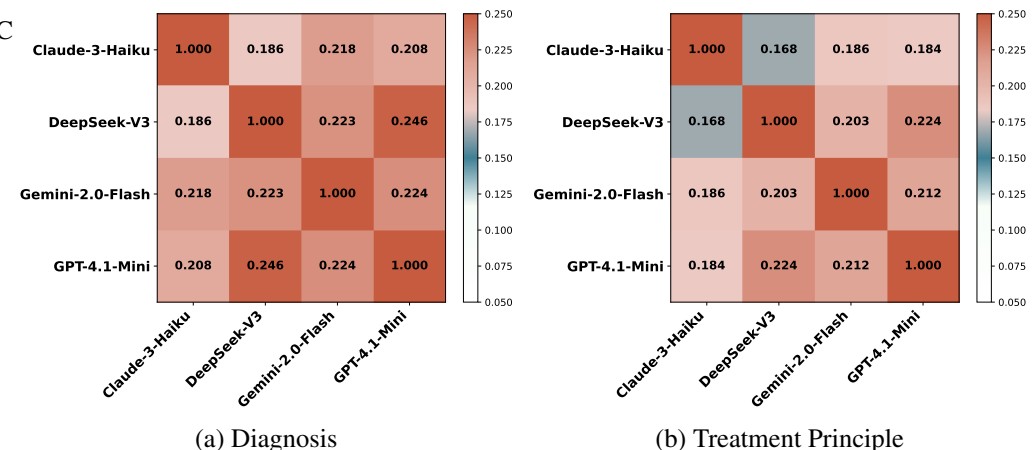

(a) Diagnosis                                    (b) Treatment Principle

Figure 12: Cross-backbone analysis. We use jaccard similarity to measure the how similar TCMA-GENT's response toward same patient with different LLM backbones.

Figure 12 presents a cross-backbone analysis of TCMAGENT using Jaccard similarity to evaluate how similarly different LLM backbones respond to the same patient. Panel (a) shows diagnosis, where pairwise similarities range from 0.186 to 0.246, indicating modest overlap in diagnostic outputs across models. Panel (b) shows treatment principle, where similarities are slightly lower, between 0.168 and 0.224, suggesting that while backbones converge somewhat on diagnostic patterns, their proposed treatment principles diverge more. Overall, the results highlight that TCMAGENT maintains only moderate consistency across different LLM backbones, with greater variability appearing in treatment planning than in diagnosis.

Table 3 summarizes the actionability scores of treatment recommendations across medication, diet, lifestyle, and overall domains for different LLM backbones integrated with TCMAGENT. `DeepSeek-v3` and `Gemini-2.0-Flash` achieve the highest scores overall, both exceeding 0.98 and showing strong performance across all domains. `GPT-4.1-Mini` performs well on diet and lifestyle but is slightly weaker on medication, while `Claude-3.0-Haiku` lags behind the other models, particularly in the medication and lifestyle categories. On average, the system

achieves high actionability across domains, with diet recommendations being the most actionable and medication the most challenging.

Table 4: Detailed result of TCMAGENT with different LLM backbones, other than consistency, other metrics are the lower the better.

| Model | Diagnosis | | | Principle | | | Treatment | | |
|---|---|---|---|---|---|---|---|---|---|
| | DE. ↓ | Cons. ↑ | Cont. ↓ | DE. ↓ | Cons. ↑ | Cont. ↓ | DE. ↓ | Cons. ↑ | Cont. ↓ |
| **GPT-4.1-mini** | 0.79 | 1.0 | 0.29 | 0.08 | 1.0 | 0.0 | 0.171 | 1.0 | 0.0 |
| **Claude-3-haiku** | 0.92 | 1.0 | 0.25 | 0.49 | 1.0 | 0.15 | 1.33 | 1.0 | 0.04 |
| **Gemini-2.0-flash** | 0.95 | 0.99 | 0.19 | 0.54 | 1.0 | 0.0 | 0.99 | 1.0 | 0.002 |
| **GPT-OSS-20b** | 1.86 | 1.0 | 0.005 | 1.98 | 1.0 | 0.002 | 1.79 | 1.0 | 0.005 |
| **LLaMA-3.3-70b** | 2.1 | 0.99 | 0.03 | 0.35 | 1.0 | 0.02 | 0.28 | 1.0 | 0.0 |
| **Deepseek-v3** | 0.79 | 1.0 | 0.29 | 0.47 | 1.0 | 0.07 | 0.82 | 1.0 | 0.001 |

Table 4 reports detailed results of TCMAGENT with different LLM backbones across the stages of diagnosis, principle formulation, and treatment. Consistency remains close to 1.0 for all models, indicating stable behavior regardless of backbone choice. However, detected error and contraindication rates vary: `GPT-4.1-mini` and `Deepseek-v3` achieve the lowest error levels overall, while `Claude-3-haiku` and `Gemini-2.0-flash` show moderate errors with occasional contraindications. In contrast, `GPT-OSS-20b` exhibits the highest error rates across all stages, and `LLaMA-3.3-70b` displays particularly high errors in diagnosis but lower errors in principle and treatment. These results highlight that while consistency is uniformly strong, the reliability and safety of outputs depend substantially on the chosen backbone.

Table 5: Detailed result of TCMAGENT on treatment recommendation.

| Model | Medication | | | Diet | | | Lifestyle | | |
|---|---|---|---|---|---|---|---|---|---|
| | DE↓ | Cons↑ | Cont↓ | DE↓ | Cons↑ | Cont↓ | DE↓ | Cons↑ | Cont↓ |
| GPT-4.1-mini | 0.4 | 1.0 | 0.0 | 0.11 | 1.0 | 0.0 | 0.003 | 1.0 | 0.0 |
| Claude-3-haiku | 1.66 | 1.0 | 0.09 | 1.38 | 1.0 | 0.010 | 0.94 | 1.0 | 0.015 |
| Gemini-2.0-flash | 1.47 | 0.99 | 0.003 | 1.15 | 1.0 | 0.002 | 0.37 | 1.0 | 0.002 |
| GPT-OSS-20b | 2.18 | 1.0 | 0.01 | 1.73 | 1.0 | 0.002 | 1.48 | 1.0 | 0.003 |
| LLaMA-3.3-70b | 0.44 | 1.0 | 0.02 | 0.02 | 1.0 | 0.0 | 0.2 | 0.99 | 0.0 |
| Deepseek-v3 | 1.23 | 1.0 | 0.001 | 0.77 | 1.0 | 0.000 | 0.45 | 1.0 | 0.000 |

Table 5 reports detailed results of TCMAGENT on treatment recommendation, broken down into medication, diet, and lifestyle. Across all backbones, consistency remains close to 1.0, indicating stable recommendations. However, detected error and contraindication vary: `GPT-4.1-mini` and `LLaMA-3.3-70b` achieve particularly low error and contraindication rates across modalities, while `Claude-3-haiku`, `Gemini-2.0-flash`, and `GPT-OSS-20b` exhibit higher error rates, especially in medication and diet. `Deepseek-v3` performs moderately well with strong safety and relatively low errors. These results highlight that although consistency is uniformly strong, backbone choice impacts the reliability and safety of generated treatment recommendations.

Table 6: Detailed result of Rreflection on treatment recommendation.

| Model | Medication | | | Diet | | | Lifestyle | | |
|---|---|---|---|---|---|---|---|---|---|
| | DE↓ | Cons↑ | Cont↓ | DE↓ | Cons↑ | Cont↓ | DE↓ | Cons↑ | Cont↓ |
| GPT-4.1-mini | | | | | | | | | |
| TCMAgent | 1.39 | 1.0 | 0.257 | 1.14 | 1.0 | 0.006 | 0.32 | 1.0 | 0.001 |
| TCMAgent + Reflection | 0.40 | 1.0 | 0.000 | 0.11 | 1.0 | 0.000 | 0.003 | 1.0 | 0.000 |
| Gemini-2.0-flash | | | | | | | | | |
| TCMAgent | 2.18 | 0.99 | 0.090 | 2.04 | 1.0 | 0.025 | 1.72 | 1.0 | 0.001 |
| TCMAgent + Reflection | 1.47 | 1.0 | 0.003 | 1.15 | 1.0 | 0.002 | 0.37 | 1.0 | 0.002 |
| LLaMA-3.3-70b | | | | | | | | | |
| TCMAgent | 2.3 | 1.0 | 0.1 | 2.07 | 1.0 | 0.007 | 1.09 | 1.0 | 0.001 |
| TCMAgent + Reflection | 0.44 | 1.0 | 0.02 | 0.02 | 1.0 | 0.0 | 0.2 | 0.99 | 0.0 |

Table 6 presents the impact of adding a reflection module to TCMAGENT across medication, diet, and lifestyle recommendations. For all backbones, reflection substantially reduces detected errors and contraindications while maintaining near-perfect consistency. For example, `GPT-4.1-mini` shows a sharp drop in errors (from 1.39 to 0.40 in medication and from 1.14 to 0.11 in diet) with contraindications eliminated. Similar improvements are observed for `Gemini-2.0-flash` and `LLaMA-3.3-70b`, where reflection consistently lowers error rates and contraindications, most notably reducing LLaMA's medication errors from 2.3 to 0.44. These results demonstrate that reflection strengthens reliability and safety in treatment recommendations without compromising consistency.

Table 7: DE and Contraindication results of Gemini, GPT, and Claude under different debate settings.

| Setting | Domain | Gemini-2.0-flash | | GPT-4.1-mini | | Claude-3-haiku | |
|---|---|---|---|---|---|---|---|
| | | DE | Contra | DE | Contra | DE | Contra |
| Debate | Medication | 1.471 | 0.0030 | 0.403 | 0.000 | 1.663 | 0.087 |
| Debate | Diet | 1.146 | 0.0010 | 0.110 | 0.000 | 1.381 | 0.010 |
| Debate | Lifestyle | 0.371 | 0.0010 | 0.003 | 0.000 | 0.941 | 0.015 |
| No-debate | Medication | 1.650 | 0.190 | 0.710 | 0.040 | 2.000 | 0.150 |
| No-debate | Diet | 1.250 | 0.000 | 0.190 | 0.000 | 1.140 | 0.010 |
| No-debate | Lifestyle | 0.800 | 0.000 | 0.000 | 0.000 | 0.870 | 0.140 |
| Aggressive | Medication | 2.510 | 0.350 | 1.960 | 0.380 | 1.780 | 0.330 |
| Aggressive | Diet | 2.770 | 0.320 | 1.580 | 0.050 | 1.500 | 0.090 |
| Aggressive | Lifestyle | 2.690 | 0.000 | 1.240 | 0.100 | 1.040 | 0.070 |
| Conservative | Medication | 1.980 | 0.050 | 1.250 | 0.000 | 1.890 | 0.320 |
| Conservative | Diet | 1.440 | 0.000 | 0.570 | 0.000 | 1.590 | 0.040 |
| Conservative | Lifestyle | 0.830 | 0.060 | 0.100 | 0.020 | 1.020 | 0.150 |

Table 7 compares detected error (DE) and contraindication rates for `Gemini-2.0-flash`, `GPT-4.1-mini`, and `Claude-3-haiku` under different debate settings across medication, diet, and lifestyle domains. Overall, the debate setting consistently reduces both errors and contraindications relative to the no-debate baseline, with GPT showing the strongest improvements, achieving near-zero contraindications across all domains. Gemini also benefits, particularly in lowering contraindications, though its error levels remain higher than GPT. Claude achieves moderate reductions but still shows relatively elevated error rates. By contrast, the aggressive setting substantially increases both errors and contraindications, while the conservative setting yields intermediate results. These findings highlight that structured debate can enhance safety and reliability, whereas overly aggressive strategies degrade performance.

Table 8: Detailed investigation on Retrieval for treatment principle.

| Model | DE ↓ | Cons ↑ | Cont ↓ |
|---|---|---|---|
| GPT-4.1-mini | | | |
| w/o RAG | 0.47 | 1.0 | 0.070 |
| w/ RAG | 0.08 | 1.0 | 0.000 |
| Gemini-2.0-flash | | | |
| w/o RAG | 0.60 | 1.0 | 0.024 |
| w/ RAG | 0.54 | 1.0 | 0.014 |
| LLaMA-3.3-70b | | | |
| w/o RAG | 1.03 | 1.0 | 0.09 |
| w/ RAG | 0.35 | 1.0 | 0.02 |

Table 8 examines the effect of retrieval on treatment principle generation across three backbones. Consistency remains perfect at 1.0 in all cases, but retrieval substantially reduces both detected errors and contraindications. For `GPT-4.1-mini`, errors drop from 0.47 to 0.08 and contraindications are eliminated, while `LLaMA-3.3-70b` shows a marked improvement from 1.03 to 0.35 in errors and from 0.09 to 0.02 in contraindications. `Gemini-2.0-flash` benefits less strongly, though retrieval still lowers contraindications. These results demonstrate that retrieval integration enhances reliability and safety without affecting consistency.

# D  EXAMPLE OUTPUT OF TCMAGENT

In this section, we showcase the some examples of diagnosis, treatment principles, and treatment recommendations generated by TCMAGENT.

| Diagnosis and Treatment Principle | Treatment Recommendation |
|---|---|
| **Retrieved Document**: 方法，就是扶正的具体方法，而发汗、吐下等方法，则是祛邪的具体方法。
（三）治疗原则
祖国医学认为："治病必求于本"（《素问·阴阳应象大论》）。本，本质、本原、根本、根源之谓。治病求本，就是在治疗疾病时，必须寻找出疾病的根本原因，抓住疾病的本质，并针对疾病的根本原因进行治疗。它 是 中医辨证论治的一个根本原则，也是中医治疗中最基本的原则。阴平阳秘，精神乃治，阴阳乖戾，疾病乃起。阴阳失调是人体失去生理状态而发生病理变化的根本原因，治疗疾病就是要解决阴阳失调偏胜偏衰的矛盾，使之重归于新的动态平衡。所以，治病求本，本者本于阴阳之谓，即治病必须追究疾病的根本原因，审察疾病的阴阳逆从，而确定治疗方法。"故凡治病者，在必求于本，或本于阴，或本于阳，知病之所由生而直取之，乃为善治。若不知根本，则茫如望洋，无可问津矣"（《医门法律·申明内经法律》）。阴阳失衡是疾病的根本矛盾。治本的基本原则就是调整阴阳，"谨察阴阳之所在而调之，以平为期"

**Diagnosis:**
### Traditional Chinese Medicine (TCM) Diagnostic Reasoning:
**Pattern Identification:**
1. Liver Qi Stagnation Persistent localized pain and swelling in the breast region correspond with stagnation of Liver Qi, as the Breast and chest area are closely related to Liver and Stomach meridians. Pain and tenderness with poor mobility and hardness reflect Qi blockage.
2. Blood Stasis: Hard texture, fixed mass with clear boundaries, and subacute nature of symptoms strongly indicate blood stasis (stagnant blood), preventing smooth flow and causing local accumulation and pain.
**Etiology and Pathomechanism:** **
1.Initial Qi stagnation (often due to emotional stress or other internal imbalance) leads to disruption of smooth Liver channel flow.
2. Liver Qi stagnation over time transforms into Liver Fire or localized heat, causing redness, swelling, and pain.
3. Prolonged stagnation results in Blood stasis and phlegm accumulation, creating a palpable hard mass.
4. Secondary bacterial infection (*Staphylococcus aureus*) corresponds with heat toxin and infectious inflammation.

**Treatment Principle:**
The treatment principle for this patient should focus on addressing both the root (本) and the branch (标) of the disease process, following the fundamental TCM principle of "treating the root cause" (治病必求于本). The identified root causes are Liver Qi stagnation transforming into heat, blood stasis obstructing the collaterals, and mild Liver Yin deficiency. The branch manifestations include localized heat-toxin infection, damp-phlegm accumulation, and inflammatory swelling.
### Treatment Principle (治则):
1. **Soothe and regulate Liver Qi**
  - To relieve Liver Qi stagnation, restore smooth flow of Qi, and prevent further transformation into Heat and Blood Stasis. This addresses the fundamental blockage and disharmony of Liver function.
2. **Clear Heat and detoxify Toxin**
  - To eliminate the localized heat toxin associated with *Staphylococcus aureus* infection, reduce redness and swelling, and resolve inflammatory manifestations.
3. **Invigorate Blood and resolve Blood Stasis**
  - To promote blood circulation, break up blood stasis causing hardness and mass formation, and facilitate the dispersal of stagnant accumulations.
4. **Transform and eliminate Phlegm and Dampness**
  - To remove turbid phlegm and dampness obstructing the channels and tissues, thus aiding in resolving the mixed echogenic lesion and fluid accumulation. | **Medication:**
Core Modified Xiao Yao San:
- Chai Hu (Bupleurum) 9g
- Dang Gui (Angelica root) 9g
- Bai Shao (White peony) 9g
- Bai Zhu (Atractylodes) 6g
- Fu Ling (Poria) 9g

Heat-Clearing & Detoxifying Group:
- Huang Qin (Scutellaria) 6g
- Jin Yin Hua (Lonicera flower) 6g
- Lian Qiao (Forsythia) 6g

Blood-Invigorating & Stasis-Resolving Group:
- Chuan Xiong (Ligusticum) 6g
- Hong Hua (Safflower) 4g ;
- Tao Ren (Peach kernel, toasted) 4g

Phlegm-Transforming & Dampness-Eliminating Group:
- Ban Xia (Pinellia, prepared) 6g
- Chen Pi (Citrus peel) 6g
- Fu Ling (Poria) 9g (already included in core formula)

**Diet:**
** Diet plan** :
- Breakfast: Millet-oat porridge with a pinch of turmeric and ginger; cooked pear slices
- Mid-morning tea: Chrysanthemum–honeysuckle infusion (warm)
- Lunch: Steamed fish with garlic and fennel; sautéed dandelion greens; barley congee
- Afternoon snack: Small bowl of mung bean soup; goji-date tea
- Dinner: Chicken soup with lotus root and lily bulb; watercress salad lightly dressed with lemon zest and olive oil
- Evening: Black sesame–silken tofu pudding

** Foods to Strictly Avoid **:
- Fried, greasy, overly spicy foods
- Alcoholic or caffeinated drinks
- Excessive raw cold salads
- Dairy products

**Lifestyle:**
1. Gentle-to-Moderate Aerobic Movement (20–30 minutes per day, 5–6 days/week)
• Start with slow-paced walking or light cycling for 15–20 minutes.
• Increase duration by 2–3 minutes each week until reaching 30 minutes.
• Goal within 2–3 weeks: brisk walking or gentle swimming.
2. Qi Gong or Tai Chi (15 minutes daily)
• Focus on Liver-meridian patterns (e.g., "White Crane Spreads Wings," "Liver Clearing Circle").
• Integrate deep diaphragmatic breathing.
3. Meridian-Point Self-Massage (10 minutes morning and evening)
• Light-to-moderate pressure on LR3 (Taichong) and LR14 (Qimen).
• Include gentle breast-adjacent stroking toward the axilla.
4. Stress Management & Sleep Hygiene
• Practice 5 minutes of seated mindfulness or guided imagery after Qi Gong.
• Maintain a fixed bedtime and wake-up time to secure 7–8 hours of restful sleep.
5. Symptom Monitoring
• Daily check of the breast lesion for changes in heat, redness, or pain.
• Pause or scale back activity immediately if local signs worsen, and report promptly. |

Figure 13: Example output of TCMAGENT.

## Diagnosis and Treatment Principle

**Retrieved Document**: 耗伤肾阴。据此清代医家叶天士提出了"务在先安未受邪之地"的防治原则。在甘寒以养胃阴的方药中，加入"咸寒"以养肾阴的药物，从而防止肾阴耗伤。

**Diagnosis:**

### Pattern Analysis and TCM Diagnosis

1. **Chronic Nature and Clinical Presentation**
- The very long duration and gradual progression indicate a **chronic internal imbalance**, excluding acute external pathogenic factors such as wind, heat, or toxic fire.
- The absence of inflammation signs (no redness, no heat, no nodules or fixed masses) excludes active **Phlegm-Heat** or **Blood Stasis heat signs**.
- Symmetrical breast enlargement without palpable masses or lymphadenopathy suggests systemic nature rather than localized tumor-like pathology.

2. **Main Pathogenic Patterns**
 **Liver Qi Stagnation transforming into Phlegm-Dampness**
 - The breasts are closely related to the Liver channel in TCM. Stagnant Liver Qi, especially over many years, can generate **Phlegm and Dampness** due to disrupted Qi flow.
 - This results in swelling/enlargement and tissue changes.
 **Spleen Qi Deficiency with Damp Retention**
 - The Spleen governs transport and transformation of fluids.
 - Deficiency here leads to accumulation of Damp and Phlegm, worsening tissue swelling and inability to resolve stagnation.
**Kidney and Liver Yin Deficiency**
 - Long-standing imbalance leads to depletion of Liver and Kidney Yin, which normally nourish and maintain tissue integrity, including breast tissue.
 - Yin deficiency may cause structural weakness contributing to tissue sagging and persistent enlargement.

## Supporting Evidence from Reports
- **Physical Exam:** No masses, no inflammation, symmetrical enlargement with mild sagging — supportive of Qi/Blood deficiency and Liver Qi stagnation without Heat or stasis.
- **Laboratory Data:** Normal blood counts, absence of infection/inflammation; dyslipidemia consistent with Spleen Qi deficiency and Damp retention pattern.

**Treatment Principle:**

1. **Soothing Liver Qi:**
Liver Qi stagnation is the initial pathogenic factor causing Qi movement disturbance, leading to the formation of Phlegm and Dampness. Promoting the free flow of Liver Qi is fundamental to prevent further accumulation of pathological fluids and relieve tissue swelling.

2. **Resolving Phlegm-Dampness:**
Phlegm and Dampness are the direct pathological substances contributing to tissue enlargement. Clearing and transforming these pathogenic factors is necessary to reduce swelling and improve local tissue status.

3. **Tonifying Spleen Qi:**
The Spleen governs fluid metabolism and transformation. Spleen Qi deficiency leads to failure of Dampness distribution and removal, exacerbating fluid retention. Strengthening Spleen Qi improves transport and transformation functions, helping to drain Dampness and reduce Phlegm formation.

4. **Nourishing Liver and Kidney Yin:**
Long-standing disease and chronic Qi stagnation have depleted Yin, leading to poor nourishment of breast tissue and structural support. Enriching Liver and Kidney Yin restores tissue integrity and prevents further degeneration.

5. **Supplementing Qi and Blood:**
Mild Qi and Blood deficiency implies inadequate nourishment and support for tissues, manifesting clinically as tissue sagging and possibly diminished vitality. Tonifying Qi and Blood supports overall health and promotes recovery of tissue function.

## Treatment Recommendation

**Medication:**
Base to Soothe Liver Qi and Resolve Phlegm-Dampness (Xiao Yao San + Er Chen Tang blend)
- Chai Hu (Bupleuri Radix) 8g
- Bai Shao (Paeoniae Radix Alba) 10g
- Ban Xia (Pinelliae Rhizoma) 6g
- Chen Pi (Citri Reticulatae Pericarpium) 6g

Tonify Spleen Qi and Improve Fluid Metabolism
- Dang Shen (Codonopsis Radix) 10g
- Bai Zhu (Atractylodis Macrocephalae Rhizoma) 8g
- Fu Ling (Poria) 8g
- Huang Qi (Astragali Radix) 6g

Nourish Liver & Kidney Yin
- Shu Di Huang (Rehmanniae Radix Praeparata) 8g
- Mai Men Dong (Ophiopogonis Radix) 6g
- Nu Zhen Zi (Ligustri Lucidi Fructus) 6g

**Diet:**
1. Soothe Liver Qi and Promote Free Flow
 • Include:
 – Fresh green leafy vegetables (e.g., spinach, kale, watercress)
 – Moderate mandarin peel (Chen Pi)
 • Limit:
 – Excessively sour fruits or strong aromatics
2. Resolve Phlegm-Dampness
 • Include:
 – Diuretic grains: barley and coix seed (job's tears)
 – Bitter winter melon and moderate adzuki beans
 • Limit:
 – Heavy beans or excessive cold, "draining" foods
3. Tonify Spleen Qi
 • Include:
 – Warmly cooked grains (millet, rice, oats) as porridges
 – Root vegetables (pumpkin, sweet potato)
 – Moderate lean protein (chicken, egg)
 • Avoid:
 – Raw salads, cold drinks
4. Nourish Liver & Kidney Yin
 • Include:
 – Moistening foods: goji berries, black sesame seeds, lily bulbs, lotus seeds
 – Hydrating fruits: pear and apple
 • Avoid:
 – Alcohol, spicy foods

5. Supplement Qi and Blood
 • Include:
 – Red dates and longan fruit in soups/teas
 – Angelica root (Dang Gui) and warm bone broth
 • Avoid:
 – Excessive caffeine or stimulants

**Lifestyle:**
- Wake between 6:00–7:00 am; lights out by 10:00 pm.
- Every 2 hours, pause for 3–5 minutes of gentle stretching or breathing.
- Sip warm water or mild ginger tea throughout the day; avoid iced drinks.
- Take short, unpressured breathing breaks or 1–2 minutes of seated Qi gathering when feeling tense.

Figure 14: Example output of TCMAGENT.

| Diagnosis and Treatment Principle | Treatment Recommendation |
|---|---|
| **Retrieved Document**: 学预防疾病思想最突出的体现。"是故已病而后治，所以为医家之法；未病而充治，所以明摄生之理"（《丹溪心法》）。未病先防旨在提高抗病能力，防止病邪侵袭。
（二）未病先防的方法
1．调养身体，提高人体抗病能力
（1）调摄精神：精神情志活动是脏腑功能活动的体现。突然强烈的精神刺激，或反复的、持续的刺激，
可以使人体气机紊乱，气血阴阳失调而发病，而在疾病的过程中，情志变动又能使疾病恶化。因此，调养
精神就成为养生的第一要务了。
中医摄生十分重视精神调养，要求人们做到"恬淡虚无"。"恬"是安静；"淡"是愉快；"虚"是虚怀若谷，
虚己以待物；"无"足没有妄想和贪求，即具有较为高尚的情操，无私寡欲，情舒畅，精神愉快，则人体的气
机调畅，气血和平，正气旺盛，就可以减少疾病的发生。
（2）锻炼身体："生命在于运动"。人体通过运动，可使气机调畅，气血流通，关节疏利，增强体质，提高
抗病力，不仅可以减少疾病的发生，促进健康长寿，而且对某些慢性病也有一定的治疗作用。

**Diagnosis:**
**Primary Pattern:**
- **Wind-Heat invasion with Heat-toxin accumulation in the Lung**
- **Phlegm-Heat obstructing the Lung**
- **Heat toxin affecting the nutritive (ying) and possibly blood (xue) levels**
- **Spleen Qi deficiency with impairment of fluid metabolism**
- **Emerging disturbance of Shen (Mind) due to internal heat and phlegm-heat blocking or damaging the Heart and Liver**

### Evidence and Reasoning from Provided Data:
1. **External Invasion of Pathogenic Factor (Wind-Heat):**
- Clinical symptoms of fever (up to 40°C), cough with difficult-to-expectorate phlegm, and pharyngeal congestion indicate an **external Wind-Heat pathogenic invasion affecting the Lung**.
- Physical exam shows moist red lips and no dryness, corresponding to relatively preserved Yin fluids but active heat.
2. **Heat-Toxin Accumulation and Phlegm-Heat Obstruction:**
- Despite antibiotic treatment, recurrent fever and worsening headache and poor mental response indicate **internal heat-toxin transformation** and **phlegm-heat retention in the Lung**.
- High inflammatory markers IL-6, SAA, and Procalcitonin reflect severe internal heat and toxicity, consistent with **Heat toxin accumulation** in TCM.

**Treatment Principle:**
1. **Clear Heat and Detoxify Toxic Heat**
- The primary pathology includes Wind-Heat invasion progressing into internal Heat-Toxin accumulation affecting Lung, Ying (nutritive), and Xue (blood) levels.
- The principle is to clear pathogenic Heat and resolve Toxic Fire to prevent deeper tissue damage and systemic impairment.

2. **Resolve Phlegm-Heat and Facilitate Lung Qi**
- Phlegm-Heat obstructing the Lung impairs respiration and fluid metabolism, contributing to cough with thick sputum and Lung Qi dysfunction.
- Treatment should clear phlegm-heat, transform and expel phlegm, and restore proper dispersing and descending function of Lung Qi.

3. **Nourish Yin and Cool Ying and Blood Levels**
- Heat toxin injures Yin fluids, especially at the Ying and Blood levels, causing disturbance of the Shen and mental symptoms.
- Treatment must nourish Yin (especially Lung and Kidney Yin), clear Ying-level heat, cool blood, and calm the mind to stabilize Shen. | **Medication:**
A. Base Formula:
– Huang Qin 3g
– Jin Yin Hua 4g
– Lian Qiao 4g
– Ban Lan Gen 2g
– Xuan Shen 2g

B. Phlegm-Heat Resolution
– Gua Lou Pi 3g
– Zhe Bei Mu 2g
– Qian Hu 2g

C. Yin Nourishment & Cooling of Ying/Blood
– Sha Shen 4g
– Mai Men Dong 4g
– Sheng Di Huang 3g

D. Spleen Qi Support & Fluid Metabolism
– Dang Shen 3g
– Bai Zhu 3g
– Fu Ling 3g

**Diet:**
1. Warm Congee Base (to strengthen Spleen Qi)
• Plain white-rice congee with a small amount of lean chicken or fish (3–4 tsp per meal)
• Stir in a few slivers of fresh ginger (1–2 very thin slices)

2. Heat-Clearing, Yin-Nourishing Add-Ins (moderate and cooked)
• Pear & lotus-seed porridge: diced pear + soaked lotus seeds, simmered in congee for 10–15 min
• Steamed lily bulbs & Chinese yam: 1 Tbsp each, added to congee or a separate small bowl of porridge
• Goji berries & black sesame: 5–8 gojis + 1 tsp sesame seeds, stirred into warm porridge at end of cooking

3. Phlegm-Heat Resolving Vegetables (cooked, not raw)
• Winter melon soup: diced winter melon + a few slices of radish, simmered with lean pork/fish

**Lifestyle:**
Phase I (Days 1–2):
- Breathing & Qi-Fostering Exercise: Once daily, 10 minutes
• Sit comfortably, inhale deeply through the nose expanding lower ribs, exhale slowly relaxing shoulders.
• Gentle shoulder rolls (5 each direction).
- Indoor Gentle Movement: Throughout the day
• Light stretching or slow playful movements in a warm, quiet room.
• Avoid prolonged sitting or immobility.
- Rest: Schedule a rest period or nap after the exercise session.
- Sinus Care: Gentle warm-saline nasal spray once per day.

Phase II (From Day 3, as fever subsides and CNS symptoms improve):
- Breathing & Qi-Fostering Exercise: Twice daily, 10–15 minutes each
• Repeat Phase I breathing routine; add gentle arm lifts to chest height.
• Mindful pacing; pause if cough intensifies.
- Controlled Outdoor Walk: Once daily, 15–20 minutes
• Choose a clean, shaded area during cooler parts of the day.
• Keep pace slow; under adult supervision; stop if tired or overheated.
- Rest: Schedule a rest period or nap after each exercise session.
- Sinus Care: Progress to very gentle warm-saline irrigation followed by a 5-minute warm compress over sinuses if tolerated. |

Figure 15: Example output of TCMAGENT.

# E PROMPTS

In this section, we showcase the prompts we used. In particular, Figure 16– 20 showcase prompts used in TCMAGENT. Figure 21– 30 showcases prompts for evaluation.

---

**Prompt for Analyst**

**System:**
You are a helpful AI assistant, collaborating with other assistants. Use the provided tools to progress towards answering the question. If you are unable to fully answer, that's OK; another assistant with different tools If you or any other assistant has the <Type of data>: **ANALYSIS** or deliverable, prefix your response with <Type of data> ANALYSIS: **ANALYSIS** so the team knows to stop."
You have access to the following tools: {tool_names}. with analyzing <Data Name> of a patient. Your analysis should make use of knowledge of Traditional Chinese medincine.

**Instruction:**
Please write a comprehensive report of the analysis in order to gain a full view of patient's information to inform other doctors. Make sure to include as much detail as possible. Provide detailed and finegrained analysis and insights that may help doctor make decisions.

---

Figure 16: Prompt for Analyst.

---

**Prompt for Diagnose**

**System:**
You are a helpful AI assistant, collaborating with other assistants.

**Instruction:**
Your goal is to use Chinese medical knowledge to diagnose your patient comprehensively.
Key focus:
- Consider each report thoroughly, focus on diagnose. Extract your evidence from each report.
- Pay attention to connection between report information I gave to you.
- I would give you following report information, use information I gave you to answer.
Basic information report: {basic_report}
Lab EXAM report: {laboratory_examination_report}
Physical EXAM report: {physical_examination_report}
Medical RECORD report: {medical_history_report}
Pathology EXAM report: {pathology_examination_report}
Imageological EXAM report: {imageological_examination_report}

Use this information to deliver your diagnostic result.

---

Figure 17: Prompt for treatment principle.

---

**Prompt of Treatment Principle**

**System**
You are a traditional Chinese medical doctor agent analyzing diagnose of patient and making treatment principal. Based on a comprehensive analysis by a team of analysts, here is a diagnose tailored for patient. This diagnose incorporates insights and knowledge of Traditional Chinese medicine. Your treatment principal should be an overview, not too specific and detailed. Leverage these insights to make an informed principal of treatment using Traditional Chinese medical knowledge.

**Instruction**
Use this diagnose as a foundation for generating your treatment principal.\n Diagnose:
{diagnose}

---

Figure 18: Prompt for diagnose.

---

**Prompt of Debate**

As a doctor prefer conservative/aggressive treatment, your role is to make a medication/diet/lifestyle plan based on treatment principal and patient's diagnose information as well as knowledge of traditional Chinese medicine.
Use these information to strengthen your arguments and challenge the opposing views. Specifically, respond directly to each point made by the aggressive analysts, countering with data-driven rebuttals and persuasive reasoning. Highlight where their caution might miss critical opportunities or where their assumptions may be overly aggressive.

Here is the treatment principal:
{treatment_principal}

Here is the diagnose result of patient:
{diagnose}

Your task is to create a compelling case for the doctor's decision of medication/diet/lifestyle plan by questioning and critiquing the aggressive/conservative stances to demonstrate why your high-reward perspective offers the best path forward.

---

Figure 19: Prompt for debate.

## Prompt for Judge

As a risk management judge and debate facilitator, your goal is to evaluate the debate between aggressive agent and conservative agent to determine the best diet decision for doctor.
Make sure you make a diet suggestion to patient.

Guidlines for decision-making:
1. **Follow Principal of treatment**: Ensure your diet suggestion follows principal of treatment.
2. **Provide Rationale**: Support your recommendation with direct quotes and conterarguments from debate.
3. **Usage Knowledge of Traditional Chinese medicine**: Your decision should base on knowledge of traditional Chinese medicine.
4. **Learn from Past information**: Make use of information such as treatment principal and diagnose **{past_memory_str}** to help me make diet suggestion.
Deliverables:
- A clear and reasonable medication suggestion.
- Detailed reasoning anchored in the debate and past reflections
---
**Analysts Debate History:**
{history}

---
Focus on reasonable insights and continuous improvement. Build on past lessons, critically evaluate all perspectives, and ensure each decision advances better outcomes.

Figure 20: Prompt for judge.

## Prompt for Evaluating Detected Error

You are a Traditional Chinese Medicine expert evaluator.

Below is the agent's output:
{response}

Identify any factual inaccuracies or logical contradictions, with respect to Traditional Chinese Medicine, within this response.

Please answer with the number of errors detected and briefly describe them.

Please respond in the following format:
<answer>NUMBER OF ERROR HERE</answer>
<reason>YOUR REASON HERE</reason>

Figure 21: Prompt for evaluating Detected Error

## Prompt for Evaluating Contraindication

You are an impartial evaluator with expertise in Traditional Chinese
Medicine.

Patient information:
{patient}
Recommendation:
{Recommendation}

Based on the patient information, does this recommendation violate any contraindications, such as
recommending herbs, medicines, or lifestyle
advice that could be harmful or inappropriate for the patient?

Please answer with one of the following:
- "No Violation" (meaning the suggestion respects all contraindications)
- "Violation" (meaning there is at least one contraindication violation)
If you answer "Violation," please briefly specify the issue.
Please respond in the following format:
<answer>YOUR ANSWER HERE</answer>

Figure 22: Prompt for evaluating Contraindication

## Prompt for Evaluating Consistency

You are a Traditional Chinese Medicine expert evaluator. Here are all responses sequentially
from the agent's pipeline:

Previous output:
{prev_content}
Current output:
{current_content}

Evaluate how well the all response logically connects and aligns with each other. Is the
connection consistent and coherent according to Traditional Chinese Medicine principles?

Please answer: "Consistent", "Inconsistent", or "Uncertain".
Response in following format:
<answer>YOUR RESULT HERE</answer>

Figure 23: Prompt for evaluating Consistency

---

# Prompt for Evaluating Coherence

You are an impartial evaluator with expertise in Traditional Chinese Medicine.

Patient information: {patient_input}

Compare the following LLM response and Agent response on recommendations with respect to coherence. Coherence means the "clarity, logical flow, and ease of understanding of the recommendation." Rate response of Agent and LLM in terms of coherenc e Make sure to use your knowledge of Traditional Chinese Medicine.

Response A (Agent):
{agent_response}

Response B (LLM):
{llm_response}

Rules:
- Output score of agent response and llm response.
- Your score should be in range of 0 to 100.
- Rate carefully, first fully understand the definition of coherence and then begin to rate.

Response in following format:
<agent_score>YOUR AGENT SCORE HERE</agent_score>
<llm_score>YOUR LLM SCORE HERE</llm_score>

Figure 24: Prompt for evaluating Coherence

---

# Prompt for Evaluating Relevance

You are an impartial evaluator with expertise in Traditional Chinese Medicine.

Patient information: {patient_input}

Compare the following LLM response and Agent response on recommendations with respect to relevance. Relevance means the "how well the suggestions address the patient's condition and needs." Rate response of Agent and LLM in terms of relevance. Make sure to use your knowledge of Traditional Chinese Medicine.

Response A (Agent):
{agent_response}

Response B (LLM):
{llm_response}

Rules:
- Output score of agent response and llm response.
- Your score should be in range of 0 to 100.
- Rate carefully, first fully understand the definition of relevance and then begin to rate.

Response in following format:
<agent_score>YOUR AGENT SCORE HERE</agent_score>
<llm_score>YOUR LLM SCORE HERE</llm_score>

Figure 25: Prompt for evaluating Relevance

# Prompt for Evaluating Completeness

You are an impartial evaluator with expertise in Traditional Chinese Medicine.

Patient information: {patient_input}

Compare the following LLM response and Agent response on recommendations with respect to completeness. Completeness means the "how comprehensive and detailed the suggestions are." Rate response of Agent and LLM in terms of completeness. Make sure to use your knowledge of Traditional Chinese Medicine.

Response A (Agent):
{agent_response}

Response B (LLM):
{llm_response}

Rules:
- Output score of agent response and llm response.
- Your score should be in range of 0 to 100.
- Rate carefully, first fully understand the definition of completeness and then begin to rate.

Response in following format:
<agent_score>YOUR AGENT SCORE HERE</agent_score>
<llm_score>YOUR LLM SCORE HERE</llm_score>

Figure 26: Prompt for evaluating Completeness

# Prompt for Evaluating Complexity

You are an impartial evaluator with expertise in Traditional Chinese Medicine.

Patient information: {patient_input}

Compare the following LLM response and Agent response on recommendations with respect to complexity. Complexity means the "how challenges and burdens a patient faces when following multiple treatment suggestions simultaneously." Rate response of Agent and LLM in terms of complexity. Make sure to use your knowledge of Traditional Chinese Medicine.

Response A (Agent):
{agent_response}

Response B (LLM):
{llm_response}

Rules:
- Output score of agent response and llm response.
- Your score should be in range of 0 to 100.
- Rate carefully, first fully understand the definition of complexity
 and then begin to rate.

Response in following format:
<agent_score>YOUR AGENT SCORE HERE</agent_score>
<llm_score>YOUR LLM SCORE HERE</llm_score>

Figure 27: Prompt for evaluating Complexity

---

## Prompt for Evaluating Correctness

You are an impartial evaluator with expertise in Traditional Chinese Medicine.

Patient information: {patient_input}

Compare the following LLM response and Agent response on recommendations with respect to correctness. Correctness means the "accuracy and alignment with Traditional Chinese Medicine principles and knowledge." Rate response of Agent and LLM in terms of correctness. Make sure to use your knowledge of Traditional Chinese Medicine.

Response A (Agent):
{agent_response}

Response B (LLM):
{llm_response}

Rules:
- Output score of agent response and llm response.
- Your score should be in range of 0 to 100.
- Rate carefully, first fully understand the definition of correctness and then begin to rate.

Response in following format:
<agent_score>YOUR AGENT SCORE HERE</agent_score>
<llm_score>YOUR LLM SCORE HERE</llm_score>

---

Figure 28: Prompt for evaluating Correctness

---

## Prompt for Evaluating Actionability

You are an impartial evaluator with expertise in Traditional Chinese Medicine.

Patient information: {patient_input}

Compare the following LLM response and Agent response on recommendations with respect to actionability. Actionability means the "how clear, practical, and feasible the treatment suggestions are for patients to understand and implement in real-world settings." Rate response of Agent and LLM in terms of actionability. Make sure to use your knowledge of Traditional Chinese Medicine.

Response A (Agent):
{agent_response}

Response B (LLM):
{llm_response}

Rules:
- Output score of agent response and llm response.
- Your score should be in range of 0 to 100.
- Rate carefully, first fully understand the definition of actionability
 and then begin to rate.

Response in following format:
<agent_score>YOUR AGENT SCORE HERE</agent_score>
<llm_score>YOUR LLM SCORE HERE</llm_score>

---

Figure 29: Prompt for evaluating Actionability

---

## Prompt for Evaluating Terminology

You are an impartial evaluator with expertise in Traditional Chinese Medicine.

Patient information: {patient_input}

Compare the following LLM response and Agent response on recommendations with respect to terminology. Terminology means the "accuracy and clarity of medical or TCM terms used in the recommendations, ensuring they follow standard conventions and are unambiguous." Rate response of Agent and LLM in terms of terminology. Make sure to use your knowledge of Traditional Chinese Medicine.

Response A (Agent):
{agent_response}

Response B (LLM):
{llm_response}

Rules:
- Output score of agent response and llm response.
- Your score should be in range of 0 to 100.
- Rate carefully, first fully understand the definition of terminology

 and then begin to rate.

Response in following format:
<agent_score>YOUR AGENT SCORE HERE</agent_score>
<llm_score>YOUR LLM SCORE HERE</llm_score>

---

Figure 30: Prompt for evaluating Terminology

## F  USAGE OF LLM

LLM are used to aid or polish writing. LLM is used to polish the draft of paragraph. Then polished paragraph will be further modified by authors.

