# OpenReview forum: "TCMAgent: A Multi-Agent Framework for General Traditional Chinese Medicine"
_ICLR.cc/2026/Conference — ICLR 2026 Conference Withdrawn Submission_

### Official Review · Reviewer_JNXN · 2025-10-22

**Soundness:** 1
**Presentation:** 2
**Contribution:** 2
**Rating:** 2
**Confidence:** 4

**Summary:**

The paper introduces TCMAgent, a multi-agent framework designed to emulate expert-level, holistic decision-making for Traditional Chinese Medicine (TCM) clinical tasks. The architecture features three key modules: (i) parallel evidence synthesis via modality-specific agents, (ii) a knowledge-grounded inference step pairing retrieved domain knowledge with diagnosis, and (iii) a deliberative recommendation module where adversarial agents debate potential actions. The framework is evaluated against several proprietary and open-source LLMs on a multi-modal TCM clinical dataset, with results showing improved safety, coherence, and interpretability versus LLM baselines.

**Strengths:**

1. The paper operationalizes a multi-agent workflow for TCM, coordinating parallelized evidence gathering, structured adversarial deliberation, and reflective learning.
2. The experimental results is thorough, with direct comparisons to strong proprietary and open-source LLMs, and with detailed metric-based benchmarking.
3. The paper includes extensive ablation studies isolating the effects of knowledge retrieval, debate, and reflection mechanisms.

**Weaknesses:**

1. The overall framework more closely resembles a meticulously orchestrated multi-step prompt chain, rather than a genuine agent architecture featuring novel learning mechanisms or reasoning structures. Its contributions are predominantly reflected in prompt engineering and domain-specific applications, rather than in innovations at the foundational model or algorithmic level.
2. There is no experiment or discussion comparing medical multi-agent framworks such as MedAgents (X Tang et al., 2023) and MDAgents (Y Kim et al., 2024) or voting-based non-agentic approaches.
3. All evaluations are conducted on a single dataset (ClinicalLab, 1,500 samples). Broader generalization, transfer to other TCM datasets, or wider clinical use cases (other diseases, Western clinical benchmarks, etc.) are not tested.
4. The case study only presents the final perfect outcome, but fails to reveal the core mechanisms claimed by the paper, such as the debate process between Aggressive debator and Conservative debator, how parallel analysis among multiple agents is integrated, and how the mechanism of reflection operates.

**Questions:**

1. How does parallel encoding and multi-agent debate affect latency and resource consumption, especially as the number of agents or rounds scales?
2. While the reflection and retrieval enhancements are empirically shown to reduce error/contraindication, the analysis of their practical limitations is superficial. For example, when historical traces are sparse or knowledge retrieval is noisy, what is the system's failure mode? There is no in-depth case study or error analysis, and failure cases are not qualitatively explored.

---

### Official Review · Reviewer_UXsh · 2025-10-31

**Soundness:** 2
**Presentation:** 2
**Contribution:** 2
**Rating:** 2
**Confidence:** 5

**Summary:**

This paper introduces TCMAGENT, a multi-agent framework designed to replicate the complex reasoning of Traditional Chinese Medicine (TCM) practitioners. The system uses parallel agents for multi-modal data synthesis, a retrieval-augmented knowledge base for diagnosis, and a collaborative, adversarial debate module to refine treatment recommendations.

**Strengths:**

The paper's primary strength lies in its architecture design that operationalizes a distributed and reflective clinical workflow.

**Weaknesses:**

My major commens are:
1. The entire evaluation relies on using GPT-4.1-mini as an automated judge. This methodology is highly questionable for a high-stakes medical domain. It is prone to self-enhancement bias (when evaluating its own backbone) and potential alignment failures.
2. The paper's own analysis reveals that medication recommendations have the lowest actionability scores. The reason given is that they "often lack precise dosage or administration instructions", which makes the generated treatment plans clinically incomplete and unsafe for real-world use.
3. The novelty is highly limited and the real-world clinical usage is also limited. The framework's effectiveness is not universal. While it improves performance for models like GPT-4.1-mini and DeepSeek-v3, it causes significant performance degradation across most metrics for LLaMA-3.3-70b and yields mixed results for Gemini-2.0.
4. The cross-backbone analysis found very low Jaccard similarity in the generated treatment recommendations. The paper frames this as diversity that is similar to real-world clinical scenario. This is a weak interpretation; such high variance more likely indicates a lack of robustness and stability in the framework's outputs.
5. The "experiential reflection mechanism" is presented as a key innovation. However, its implementation is described vaguely as retrieving "deliberation traces from historical cases". The paper fails to specify how these traces are retrieved, represented, or used by the Judge Agent, making this contribution difficult to assess or reproduce.

**Questions:**

Please refer to my comments above.

---

### Official Review · Reviewer_mJyW · 2025-11-02

**Soundness:** 2
**Presentation:** 3
**Contribution:** 2
**Rating:** 4
**Confidence:** 3

**Summary:**

The paper presents TCMAGENT, a multi-agent framework for Traditional Chinese Medicine (TCM) decision-making. It processes six types of clinical inputs through specialized agents, integrates knowledge via a TCM RAG database, and generates treatment plans through an internal debate between aggressive and conservative strategies, overseen by a reflective judge. Tests on the ClinicalLab dataset (1,500 cases) show that TCMAGENT consistently improves relevance, accuracy, and safety over standard LLMs. Key contributions include: (1) a distributed multi-modal reasoning design; (2) a debate–judge–reflection mechanism for explicit trade-offs; and (3) empirical evidence of reduced errors and contraindications.

**Strengths:**

1. Clear architecture (Fig. 1) with separate analysis, diagnosis, and debate stages for interpretability and ablation.
2. Retrieval and reflection/debate components (Figs. 4–6; Tables 6–8) consistently reduce errors and contraindications.
3. Code and prompt details (Appendix E) are publicly available.

**Weaknesses:**

1. The evaluation relies almost entirely on an LLM-as-judge framework, using GPT-4.1-mini as the evaluator (Sec. 4.1) while also including it as a backbone model in the system (Table 1), introducing potential coupling and bias. Moreover, no human expert evaluation is provided to validate the clinical accuracy, safety, or reliability of the judging criteria.

2. Baseline fairness is questionable: $\textbf{while baselines are given inputs sequentially “to mimic a clinical scenario”}$,
$\textbf{TCMAGENT receives all modalities simultaneously (Sec. 4.1)}$, conflating the benefits of agent collaboration with those of richer input access. As a result, it remains unclear whether the observed gains stem from the multi-agent design or improved information packaging.

3. The paper overstates its novelty and generalization. Although it claims to provide the “first empirical evidence” that distributed, deliberative agent architectures outperform monolithic models in complex medical settings (pp. 1–3), it does not include direct comparisons with previously cited agent-based medical systems such as $\textbf{ClinicalAgent}$ [1] or $\textbf{MedAgents}$ [2], nor with competitive single-agent reasoning baselines. These omissions weaken the evidential basis for its broad claims, which should be either narrowed or supported by more comprehensive comparisons.

4. The paper’s use of the term “multi-modal” is ambiguous. All inputs appear to be textual summaries, such as written “imageological exam” reports, rather than raw images or physiological signals. Since no vision or multi-modal model is actually employed, the work does not fully support its claim of integrating parallel multi-modal evidence.

[1] Yue, Ling; Xing, Sixue; Chen, Jintai; Fu, Tianfan. ClinicalAgent: Clinical Trial Multi-Agent System with Large Language Model-based Reasoning. In Proceedings of the 15th ACM International Conference on Bioinformatics, Computational Biology and Health Informatics (ACM BCB 2024), article 11, 2024.

[2] Tang, X., Zou, A., Zhang, Z., Li, Z., Zhao, Y., Zhang, X., Cohan, A., & Gerstein, M. (2024). MedAgents: Large Language Models as Collaborators for Zero-shot Medical Reasoning. In Findings of the Association for Computational Linguistics: ACL 2024, 599-621.

**Questions:**

N/A

**Details Of Ethics Concerns:**

They generate clinical-style treatment recommendations and evaluate “safety” only with LLM-as-judge (no clinician review or rule-based checks), creating potential patient-harm risk if adopted or replicated. The KB/memory sources and leakage controls are under-specified.

---

### Official Review · Reviewer_zYFR · 2025-11-09

**Soundness:** 2
**Presentation:** 3
**Contribution:** 2
**Rating:** 2
**Confidence:** 4

**Summary:**

The work proposes a multi-agent design specifically for Traditional Chinese Medicine clinical decision-making. The system distributes reasoning across specialized agents for data analysis, diagnosis, and treatment deliberation, followed by a reflection phase to refine reasoning. The authors claim that this agent-based workflow better handles multi-modal patient information and conflicting therapeutic principles in TCM. Experimental results on a multi-modal TCM dataset show performance improvements over monolithic LLM baselines in terms of safety, coherence, etc.

**Strengths:**

- The presentation is clear.
- System design (figure 1) is comprehensive

**Weaknesses:**

- The pipeline in Figure 1 and the motivation in the introduction (see below) seem to be generally applicable to many other clinical tasks as well. I fail to understand why the authors study Traditional Chinese Medicine (TCM) specifically.
```
As a medical system practiced for millennia, TCM’s efficacy hinges on holistic diagnosisderived from heterogeneous patient data (Yue et al., 2024b; Ma et al., 2021; Wang et al., 2023). Itsglobal relevance, especially for chronic and complex conditions, underscores the need for compu-tational frameworks capable of mastering this form of reasoning (Zhang et al., 2023; Zhuang et al.,2025). Yet the core cognitive tasks of TCM—synthesizing multi-modal evidence and deliberatingover conflicting therapeutic principles—remain beyond the reach of conventional AI architectures(Zhang et al., 2025).
```
- While the entire system might be novel in that it's a specific workflow for analyzing information, each separate component is not novel, as they have been proposed or used in previous works. Therefore, I am not sure about the novel part of this work. It seems to make it more like an engineering work.

- The scope seems to be limited, as the work doesn't consider cases where there are missing data. Specifically, in
```
In this study, we leverage data from ClinicalLab (Yan et al., 2024) with their permis-sions, which contains 1,500 examples with features including patients’ medical histories, laboratory examinations, physical examinations, imaging studies, demographic information, and pathological assessments.
```

What if there's no patient medical history data?

- This is connected to the first point. In the experiments, the evaluation is a bit limited. It seems the pipeline, except for the task-specific knowledge part, can be applied to other clinical diagnosis processes.

- How do the authors know the treatment is the only ground truth? In
```
We leverage LLM-AS-JUDGE (Gu et al., 2024) to evaluate outcomes due to datascarcity of evaluating agent framework in TCM domain.
```
It indicates that there's only one ground truth, perhaps from the recommended treatment from humans?

**Questions:**

See weakness

---

### Note · Authors · 2025-11-12

I have read and agree with the venue's withdrawal policy on behalf of myself and my co-authors.